

# *De-novo* whole genome assembly of the orange jewelweed, *Impatiens capensis* Meerb. (Balsaminaceae) using nanopore long-read sequencing

Sudhindra R. Gadagkar[1,2,3], J. Antonio Baeza[4,5,6], Kristina Buss[7] and Nate Johnson[1,2,3,8]

[1] Biomedical Sciences Program, Midwestern University, Glendale, Arizona, United States of America
[2] College of Veterinary Medicine, Midwestern University, Glendale, Arizona, United States of America
[3] Arizona College of Osteopathic Medicine, Midwestern University, Glendale, Arizona, United States of America
[4] Departamento de Biología Marina, Universidad Católica del Norte, Coquimbo, Chile
[5] Department of Biological Sciences, Clemson University, Clemson, South Carolina, United States of America
[6] Smithsonian Marine Station at Fort Pierce, Fort Pierce, Florida, United States of America
[7] Bioinformatics Core, Arizona State University, Tempe, Arizona, United States of America
[8] Center for Biology and Society, Arizona State University, Tempe, Arizona, United States of America

Corresponding authors
Sudhindra R. Gadagkar,
sgadag@midwestern.edu
J. Antonio Baeza,
jbaezam@clemson.edu

## ABSTRACT

The plant family Balsaminaceae comprises only two genera, and they are a study in contrasts. While *Impatiens* is an impressively prolific genus, with over 1,000 species and more being discovered each year, its sister genus, *Hydrocera*, has one solitary species, *H. triflora*. The two genera also differ in geographic distribution and habitat type (*Impatiens* species are widely distributed in much of the Old World and N. America, while *H. triflora* is confined to wetlands specific to S. India, Sri Lanka, and SE Asia). Other contrasting features include plant habit, habitat, floral architecture, mode of seed dispersal, and a host of other traits. The family Balsaminaceae is therefore an excellent model for studying speciation and character evolution as well as understanding the proximal and evolutionary forces that have driven the two genera to adopt such contrasting evolutionary paths. Various species of the *Impatiens* genus are also commercially important in the ornamental flower industry and as sources of phytochemicals that are of medicinal and other commercial value. As a preliminary step towards studying the genomic basis of the contrasting features of the two genera, we have sequenced and assembled, *de novo*, the genome of an iconic *Impatiens* species from N. America, namely *I. capensis*, and report our findings here.

# INTRODUCTION

There has been an explosive radiation in the plant genus *Impatiens* Linnaeus (Balsaminaceae) in the last 5–6 million years (*Janssens et al., 2009*), with over 1,000 extant species distributed in diverse habitats from wet and montane forests of the tropical and subtropical regions of the Old World to temperate regions of Asia, Europe, and N. America (*Grey-Wilson, 1980b*). Indeed, new species of *Impatiens* are being discovered regularly (*e.g.*, *Xia et al., 2019*; *Mohan et al., 2020*; *Gu et al., 2021*; *Saravanan & Kaliamoorthy, 2022*). While there are larger genera (*e.g.*, *Astragalus*), *Impatiens* is particularly interesting as *Hydrocera* Wight & Arnott, the only other genus in the family, has only one species, *H. triflora*. This solitary species appears to be relatively evolutionarily static and is endemic to specific wetlands of S. India, Sri Lanka, and SE Asia. In contrast, *Impatiens* species are diverse in almost every trait. With life cycles ranging from annual to perennial, they can be herbaceous, woody shrubs, or even have a scapigerous habit. They can be found in montane rainforests, damp rock crevices, moss covered trees, dry deciduous forests, and roadside ditches, at altitudes ranging from sea level to 4,000 m. While most are terrestrial, some are epiphytic. *Impatiens* species also exhibit vast differences in various morphological characters.

This difference in evolutionary lability between its two genera makes *Balsaminaceae* an excellent model for studying speciation. Furthermore, the extremely high trait divergence among *Impatiens* species adds to the utility of this family in understanding the role of character evolution in speciation. Other features of interest in the *Impatiens* genus in the context of reproduction and speciation are: (1) it is mainly distributed in the Old World tropics and subtropics (*Janssens et al., 2006*), indicating its relatively recent origin and radiation, (2) it exhibits an enormous range in chromosome number ($2n = 6$–66), and with chromosome numbers that are not multiples of six (*e.g.*, 16, 20, *etc.*), (3) the ability to produce cleistogamous (closed) flowers and/or chasmogamous (open) flowers in the same plant (*Schemske, 1978*), indicating an elegant adaptation to the vagaries of the environment, and (4) aggressively invasive tactics, as seen for example in the Czech Republic where a single species, *I. glandulifera*, has successfully established itself in more than half of the river banks by "bribing" pollinators with nectar that is more nutritious and rewarding than that of any native plant in its habitat (*Chittka & Schurkens, 2001*). Despite these interesting features and the potential to be an exemplar for furthering our understanding of the concept of speciation (*May & Moore, 2016*; *Wiens, 2004*), this genus has surprisingly not been the subject of intensive study by ecologists and evolutionary biologists.

Finding an explanation for such stark contrasts between the two genera has been an open question in the field for more than 100 years (*Grey-Wilson, 1980a*; *Janssens, Smets & Vrijdaghs, 2012*). There has been much speculation in the literature about the reasons, mostly confined to an ecological perspective, due to the lack of molecular approaches for much of the 20[th] century. However, since even whole genome sequencing is a routine matter today, we believe that this approach needs to be invoked in comparative studies to unlock possible answers to these interesting questions concerning these two genera.

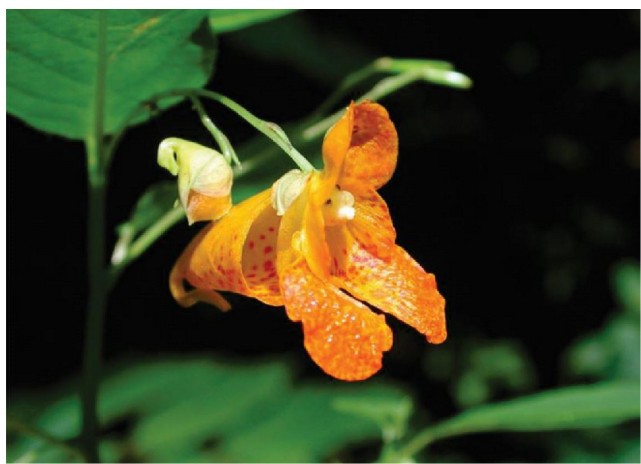

**Figure 1** *Impatiens capensis* Meerb. (Orange Jewelweed), an iconic species of the genus from N. America. Photo credit: Fritz Geller-Grimm, CC BY-SA 2.5, https://commons.wikimedia.org/w/index.php?curid=1503660.               

Apart from its importance in evolutionary studies, there is also a practical utility in genomic studies of this family. For example, *Impatiens* plants contain anti-inflammatory and fungicidal compounds such as naphthoquinone (*Li et al., 2015*; *Wang & Lin, 2012*) and several other medicinal compounds (*e.g.*, *Kim et al., 2015*; *Paun et al., 2016*; *Sun et al., 2015*). They also can be useful in making dyes and degrading toxic ones (*Roy, Ghosh & Sarkar, 2017*). Thus, the potential clearly exists for several medicinal and other commercial products from these under-studied plants. However, *Impatiens* species and hybrids are perhaps best known as highly valued ornamental flowering plants, contributing millions to the US economy (*Weining, Zhanao & Palmateer, 2015*). This industry will benefit from the systematic domestication of commercially important *Impatiens* species using molecular and genomic approaches, as in the case of the rose (*Raymond et al., 2018*). The urgency for domestication is underscored by the devastation that the industry has faced due to the so-called "downy mildew" epidemic caused by the oomycetes, *Plasmopara obducans* (*Farr & Rossman, 2020*) and *P. destrucor* (*Gorg et al., 2017*). There has been some effort in understanding the disease from the perspective of the genetics of the pathogen (*e.g.*, *Salgado-Salazar et al., 2018*). As far as the genus *Impatiens* itself is concerned, the chloroplast genomes of twelve species have been assembled recently (*e.g.*, *Luo et al., 2021*) and comparative leaf transcriptomics has been done on cultivars that differ in resistance to downy mildew (*Bhattarai et al., 2018*; *Peng et al., 2021*).

Here, we report the *de novo* sequencing and assembly of the nuclear genome of an iconic N. American species, *Impatiens capensis* (Fig. 1), as a preliminary step toward the comparative analysis of species of this genus and that of the sole congeneric species, *Hydrocera triflora*. Such a comparative study should help shed light on the genetic basis of the mode of speciation in the genus *Impatiens*, the great diversity in geographical distribution, habitat, and various other traits and contrast it with the evolutionarily static *H. triflora*. Our genomic study should also serve as a step toward the domestication of

*Impatiens* species for improving traits of commercial value, including those relating to the susceptibility/resistance to the downy mildew.

## MATERIALS AND METHODS

### Sampling and long read sequencing of *Impatiens capensis*

Tissue (leaf) samples from a specimen maintained at the Kemper Missouri Native Shade Garden, (Missouri Botanical Garden, Missouri, USA; Accession number 2001-1051) was obtained by one of the authors (SRG) and shipped to Dovetail Genomics (Scotts Valley, CA, USA) for sequencing. (While the sequencing and Hi-C refined assembly was outsourced to Dovetail, the bioinformatics for the annotation and assembly QC/ summaries were done in-house by the authors). Approximately 0.98 g of plant tissue was obtained, and genomic DNA (gDNA) extracted from the sample using the Nanobind Plant Nuclei Big DNA Kit (Circulomics Inc., San Diego, CA, USA), following the protocol detailed in *Workman et al. (2018)* that was specifically developed to obtain high molecular weight (HMW) gDNA from recalcitrant plant species for long read sequencing. (We would like to note here that multiple attempts at whole genome sequencing of *I. capensis* using PacBio technology by Dovetail proved unsuccessful. We understand from them that this was likely due to the unexpected and repeated premature separation of the polymerase from the library). A total of 11,200 ng of HMW gDNA was extracted from the sample, determined by measuring gDNA concentration with a Qubit 4 Fluorometer (Thermo Fisher Scientific, Waltham, MA, USA). Next, 100 ng of the DNA was used for library preparation, using the 1D library preparation protocol (Oxford Nanopore Technologies, Oxford, England), following the manufacturer's instructions. The library was sequenced for 48 h. on a Minion portable device with a MinION R9 flow cell (FLOMIN106, R9.4 chemistry) (Oxford Nanopore Technologies, Oxford, England), controlled with the software MiniKnow. The raw signal (FAST5 files) was basecalled using the software Guppy, version 4.0.1 (Oxford Nanopore Technologies, Oxford, UK) and a total of 3,311,362 sequencing reads in FASTQ format were obtained (available in the short read archive (SRA) repository at GenBank: accession number SAMN30713540).

### Genome assembly

Adapters were trimmed from the read ends and reads with internal adapters were split into two using the software Porechop (https://github.com/rrwick/Porechop). Next, the program fastp (*Chen et al., 2018*) was used to quality-filter the reads and retain only those sequences with Q-score ≥5. *De novo* genome assembly of *I. capensis* and decontamination were done using the pipeline wtdbg2 (*Ruan & Li, 2020*) and BlobTools (*Laetsch & Blaxter, 2017*), respectively. The contigs were further assembled into scaffolds by Dovetail using their proprietary HiRise software (*Putnam et al., 2016*) in conjunction with Hi-C data. The quality and completeness of the genome assembly were assessed using the programs QUAST v 2 (*Gurevich et al., 2013*) and BUSCO, version 5.2.2 (*Manni et al., 2021*), comparing the assembly to the eudicots_odb10 database (2020-09-10, number of genomes: 31, number of BUSCOs: 2,326).

### Genome annotation

We used the platform Genome Sequence Annotation Server (GenSAS—*Humann et al., 2019*) for the discovery and annotation of repetitive elements, protein coding genes, and other non-coding features in the newly assembled genome of *Impatiens capensis*. For the annotation of the repetitive elements, we used *de novo* prediction and homologous comparison (of the assembled contigs against the database, Repbase—(*Bao, Kojima & Kohany, 2015*), using the programs, RepeatModeler 2.0.1 (*Flynn et al., 2020*) and RepeatMasker 4.1.1 (http://www.repeatmasker.org/), respectively. Lastly, we discovered microsatellites (simple sequence repeats, SSR) using the SSR_finder script available in GenSAS.

We did the structural annotation of protein-coding genes *ab initio* using the programs Augustus 3.4.0 (*Stanke et al., 2008*), BRAKER2 2.1.5 (*Brůna et al., 2021*), GeneMarkES 4.48 (*ibid.*), GlimmerM 2.5.1 (http://ccb.jhu.edu/software/glimmerm/), and SNAP (release 11/29/2013—http://korflab.ucdavis.edu/software.html), using default parameters. Next, we produced a non-redundant reference gene set with the software EvidenceModeler 1.1.1 (*Haas et al., 2008*). We used tRNAscan-SE 2.0.7 (*Chan Patricia et al., 2021*) and RNAmmer 1.2 (*Lagesen et al., 2007*) to predict tRNAs and rRNA genes, respectively. For functional annotation of the non-redundant reference gene set, we compared the predicted proteins in the genome of *I. capensis* against the NCBI refseq plant (Protein) using BLAST+ (blastp) 2.12.0 (*Camacho et al., 2009b*) and DIAMOND 2.0.6 (*Buchfink, Xie & Huson, 2015*). Any additional homology searches were done against other four databases available in GenSAS using InterProScan 5.53-87.0 (against the InterPro database—(*Jones et al., 2014*), Pfam 1.6 (against the Pfam database—(*Finn et al., 2014*), TargetP 2.0 (*Armenteros et al., 2019a*) and SignalP 5.0 (*Armenteros et al., 2019b*)).

### Gene ontology based annotation

Interproscan (*Jones et al., 2014*), BLAST (*Camacho et al., 2009a*), and DIAMOND (*Buchfink, Reuter & Drost, 2021*) functional annotations were used to identify GO (*Ashburner et al., 2000*; *Carbon et al., 2021*) terms corresponding to predicted genes in the *I. capensis* genome. All genes with GO predictions were extracted from the complete Interproscan output file. Top alignment hits from BLAST and DIAMOND searches were cross-referenced with the OrthoDB (*Zdobnov et al., 2021*) database to find the corresponding GO terms for each predicted gene. The list of unique GO terms was weighted by taking the number of unique annotations found in the genome for each term, removing duplicates found by multiple tools, and submitting to Revigo's online service for association clustering with the default SlimRel redundancy reduction algorithm using a threshold value of 0.7. Revigo's further reduced treemap dataset, using a threshold of 0.1, was visualized with an adaptation of the open-source python package CirGO (*Kuznetsova et al., 2019*). For each of the three GO categories, the CirGO treemap reduction shows major clusters of associated terms along with their largest subclusters. Each cluster takes on the name of its primary, or representative, component, with the exception of the cellular component category "cytoplasm" that was manually named to represent a more accurate primary component instead of the automatically assigned name "ribosome". Alterations to

the code allowed the percentages to reflect the number of genes belonging to each cluster rather than the number of GO terms.

# RESULTS AND DISCUSSION

## *De novo* genome assembly using long reads

The nanopore ONT sequencing run generated a total of 3,311,362 reads (17.264 Gbp, read N50: 13,385 bp, longest read: 142,336 bp, mean read length: 5,213 bp, mean read quality (Q-score): 9.3, median read length: 2,116 bp, median read quality: 9.1). QC steps followed using porechop and fastp resulted in a total of 3,311,956 "clean" reads. Summary details regarding the assembly can be found in Fig. S1, which gives a BlobPlot of the assembly, with the base coverage of the sequence on the Y-axis and the GC content on the X-axis, along with the corresponding histograms. Sequences are represented by circles in the scatterplot that are colored by taxonomic group and the diameter of each circle is proportional to the sequence length. The count, span, and N50 values are given for each taxon in the legend. BlobTools QC indicated relatively low contamination in the assembled genome with Proteobacteria, Ascomycota, Arthropoda, Chordata, Bactereoidetes, and Chlorophyta (Fig. S1), which were filtered out from the assembly.

The circular snail plot describing the assembly statistics for the *de novo* genome sequence of *I. capensis* is shown in Fig. 2 and summarized in Table 1. The table shows the statistics of the *de novo* assembled genome after adapter-trimming, reads cleaning, and decontamination. It comprised 8,612 contigs that were a total of 725,987,335 bp in length, with N50 of 848,459 bp, and N90 of 29,117 bp. The assembled genome size of *I. capensis* in our study is slightly smaller than the 817 Mb reported for the same species when estimated using flow cytometry (*Bai et al., 2012*), but is well within the observed range reported for other congeneric species (range: 414 Mb in *I. pallida*—(*ibid.*) to 3,188 Mb in *I. omeiensis* (*Vesely et al., 2012*)).

The contigs were further assembled into scaffolds by Dovetail using their proprietary HiRise software (*Putnam et al., 2016*) in conjunction with Hi-C data, which accounted for 574,941,529 bp of the genome (or 79.19% of the genome length), as shown in Table 1 (right column). The N50 and N90 statistics for the scaffold-level assembly were 32,415,000 and 85,265 bp, respectively. The total number of scaffolds assembled from the 8,612 contigs was 2,594, out of which, 19 were greater than 1 mb in length. There was a slight change in the GC content in the scaffold-level assembly, likely due to the 20% of the genome that was not assembled into scaffolds.

Another assessment of the quality of the genome assembly is in terms of the extent to which a set of orthologous genes that are universal to a given taxonomic clade, known as BUSCOs (*Manni et al., 2021*), can be identified. The *I. capensis* genome was benchmarked using the set of 2,326 BUSCOs in the clade, Eudicota. Table 1 shows the numbers (and the corresponding percentages) of these BUSCOs identified under various categories by the contig and scaffold assemblies. Even though the BUSCO statistics are relatively low, they are impressive considering that the sequencing and assembly were entirely *de novo*, with a divergence time between *Impatiens* and the reference genome of tomato as high as 108 million years (*Kumar et al., 2022*).

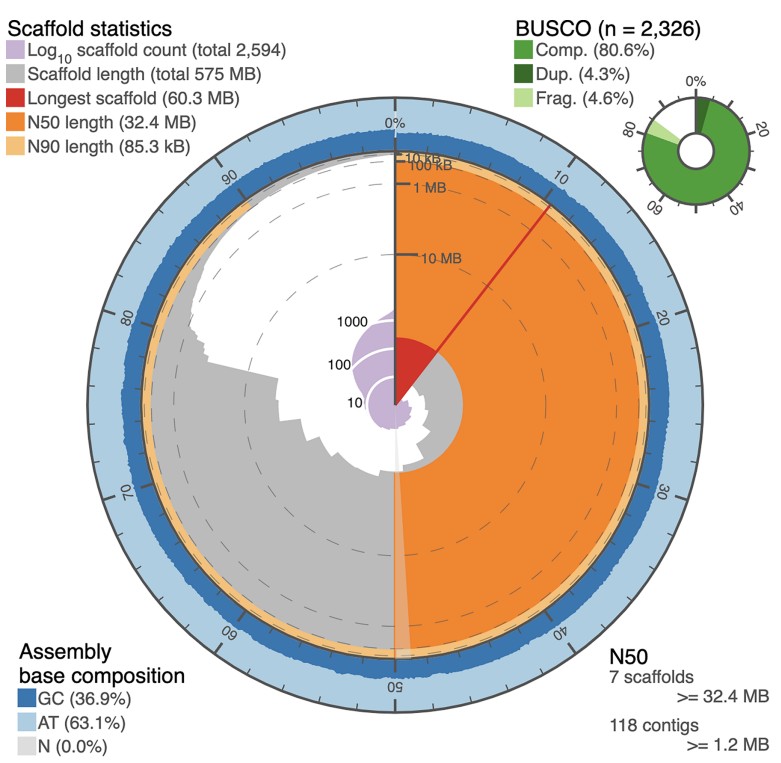

**Figure 2 Snail plot of the _I. capensis_ genome assembly.** This snail plot depicts metrics for the final scaffold-level assembly of the _Impatiens capensis_ genome. A basic legend is provided in the top left corner. The inner purple rings show the scaffold count, beginning with the largest scaffold at the black starting line and increasing as the plot proceeds clockwise to include all 2,594 scaffolds. The light grey rings plot the total length of the assembly as each additional scaffold is subtracted (when considering the plot clockwise from the vertical black line); this level of the plot shows the drastic drop in the assembly between the largest 13 scaffolds and the remaining >2,000. The red wedge highlights the longest scaffold, while the dark and light orange rings show the length of the scaffold at which 50% and 90% of the total assembly is represented, respectively. The annotation in the bottom right corner complements these orange rings with quantitative information about the number of scaffolds or contigs needed to reach 50% of the total assembly length. The outer blue rings depict the GC content over the assembly, corresponding to the scaffolds in the grey ring. The overall GC content of the assembly is provided in the bottom left corner. Finally, the top right corner provides BUSCO statistics numerically and as a pie chart. These statistics approximate the completeness of the genome using Eudicot marker genes. This plot was generated using BUSCO statistics calculated as described in _Manni et al. (2021)_ (Pages 4647–4654), and using open-source scripts accessed at https://github.com/rjchallis/assembly-stats.

An alternative graphical view of the genome assembly was provided by Dovetail Genomics in their HiRise scaffolding report in the form of a "link density histogram" (Fig. 3A), which plots the position of each mate pair (the two ends of the sequenced DNA fragments) with the position of the corresponding sequence in the assembly. Although the diploid number of chromosomes in _I. capensis_ is known to be 20 ($n = 10$) from cytological observations (_Zinov'eva-Stahevitch & Grant, 1984_), our assembly appears to have converged on 13 large scaffolds (Fig. 3A). Accordingly, we present the lengths of these 13 scaffolds (as well as the 14[th] scaffold—only to demonstrate the dramatic drop in length) in Table 2 and a circular view of these 13 scaffolds (which constitute most of the genome; Fig. 2) in the form of a Circos plot with five tracks (Fig. 3B), created using the circlize
**Table 1** Assembly statistics and BUSCO analysis.

| | Contig-level assembly | Scaffold-level assembly |
| --- | --- | --- |
| **Assembly statistics** | | |
| Total length (bp) | 725,987,335 | 574,941,529 |
| L50 & N50 (bp) | 187 & 848,459 | 7 & 32,415,000 |
| L90 & N90 (bp) | 2,660 & 29,117 | 275 & 85,265 |
| Largest contig/scaffold | 11,650,223 | 60,344,999 |
| Number of contigs/scaffolds | 8,612 | 2,594 |
| Number >1 KB/1 MB | 8,604 | 19 |
| GC content (%) | 37.23 | 36.92 |
| **BUSCO Eudicots (2,326 genes)** | | |
| Complete BUSCOs | 1,692 (72.7%) | 1,875 (80.6%) |
| Complete single-copy BUSCOs | 1,577 (67.8%) | 1,775 (76.3%) |
| Complete duplicated BUSCOs | 115 (4.9%) | 100 (4.3%) |
| Fragmented BUSCOs | 197 (8.5%) | 108 (4.6%) |
| Missing BUSCOs | 437 (18.8%) | 343 (14.8%) |

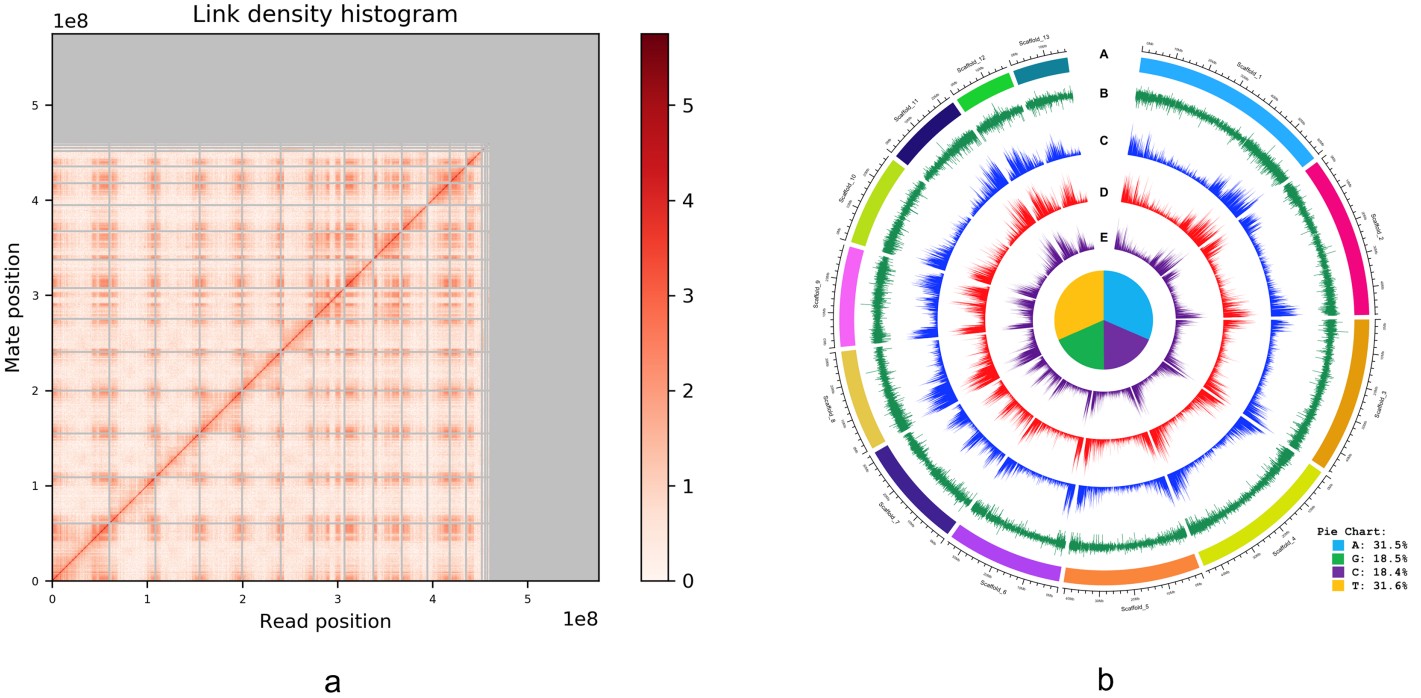

a

b

**Figure 3** **(A) Dovetail Genomic's Link Density Histogram of the _I. capensis_ genome assembly showing 13 of the largest scaffolds. (B) Circos plot of the _I. capensis_ genome with five tracks.** (B) This Circos plot depicts the 13 longest scaffolds in the final assembly, representing almost 80% of the total assembly length. In ring A, each scaffold is color-coded and labeled by name and length; ring B shows the GC content across each scaffold; ring C shows the density of annotated CDS across each scaffold; ring D shows the density of annotated genes across each scaffold; and ring E shows the density of annotated exons across each scaffold. The pie chart in the center shows the percentage of each nucleotide base in the scaffold-level assembly. This plot was created using the _circlize_ package in R as described in _Gu et al. (2014)_, and the overall GC percentage was calculated using BBtool's assembly statistics package available at https://sourceforge.net/projects/bbmap/.

**Table 2 Lengths of the largest 14 scaffolds.**

| Name | Length |
| --- | --- |
| Scaffold_1 | 60,344,999 |
| Scaffold_2 | 48,197,478 |
| Scaffold_3 | 46,397,390 |
| Scaffold_4 | 44,824,343 |
| Scaffold_5 | 40,719,996 |
| Scaffold_6 | 34,563,004 |
| Scaffold_7 | 32,415,105 |
| Scaffold_8 | 30,146,749 |
| Scaffold_9 | 29,863,999 |
| Scaffold_10 | 27,261,718 |
| Scaffold_11 | 23,126,893 |
| Scaffold_12 | 17,338,137 |
| Scaffold_13 | 16,560,492 |
| Scaffold_14 | 3,374,456 |

package in R as described in *Gu et al. (2014)*. Interestingly, all gene-related tracks showed elevated values at both edges of each scaffold—a curious feature that appears to be true in other plants as well, such as pineapple (*Yow et al., 2022*), maize (*Lin et al., 2021*), and cotton (*Peng et al., 2022*). Unsurprisingly, there appears to be some associated increase in the GC content at the scaffold edges as well. The figure also shows the composition of the four nucleotides, depicted in the form of a pie chart in the center of the Circos plot.

### Genome annotation of *I. capensis*

The non-redundant reference gene set in the nuclear genome of *I. capensis* contained a total of 26,921 protein-coding genes and 79.10% ($n = 21,294$), 79.27% ($n = 21,340$), 61.77% ($n = 16,630$), 80.97% ($n = 21,799$), and 92.22% ($n = 24,826$) of these genes with structural annotation matched entries in the NCBI refseq plant (Protein), InterPro, Pfam, SignalP, and TargetP databases, respectively. As expected, there were multiple annotations per protein coding gene. A total of 399 rRNA genes were annotated using RNAmmer, and 978 tRNAs were detected by tRNAscan-SE.

Table S1 shows the annotation of the *I. capensis* genome with respect to repeat elements. It lists the number and length for each type of repeat element found, along with the percentage of the genome covered by each element. The analyses conducted in the platform GenSAS indicated that 71% (513,815,555 bp) of the genome was composed of repetitive elements, of which, 19.50% were not annotated. Considering only classified repetitive elements (80.50%), the most abundant were retroelements LTR Gypsy/DIRS1 (24.27%) and LTR Ty1/Copia (22.25%), which were considerably more common than the DNA transposons hobo-Activator (0.12%) and Tourist/Harbinger (0.12%).

Low complexity DNA Satellites ($n = 156$; 0.001%) and rolling circles ($n = 444$; 0.07%) were much less common. The assembly also contained a total of 365,658 microsatellite sequences, most of them dinucleotides ($n = 119,744$).

### Gene ontology based annotation

Among the three sources (Interproscan, BLAST, and DIAMOND), 22,773 genes out of the 26,921 genes included in the GFF file for the assembly received from Dovetail Genomics, were annotated in some way (Table S3), representing 84.6% of the predicted genes (Fig. S2A), although not all of them had GO annotations. In total, 19,671 unique predicted genes could be mapped to at least one GO term, representing 86.4% of the annotated genes and 73.1% of the predicted genes. Figures S2A–S2C show the overlap among the three sources of annotation of the predicted genes, the genes with at least one GO prediction, and those mapped to unique GO terms, respectively. Altogether, 3,466 unique GO terms mapped to these 19,671 predicted genes. Using Revigo (*Supek et al., 2011*), we removed redundant GO terms to obtain 3,167 unique terms, of which, 424, 1,144, and 1,599 were found to belong to the Cellular Component, Molecular Function, and Biological Process categories, respectively. The remaining are obsolete, duplicated, or otherwise absent from the GO ontology used by Revigo. Summing annotations from all three tools, these GO terms represent a total of 65,211 molecular function annotations corresponding to 20,046 unique genes, 28,391 biological process annotations corresponding to 12,696 unique genes, and 13,030 cellular component annotations corresponding to 7,851 unique genes.

The number of gene annotations for the genome as a whole is larger than the number of genes in the genome as it includes multiple annotations made for each gene across all three GO namespaces; each gene annotation represents a unique combination of predicted gene and predicted GO annotation.

Figures 4A–4C are two-level pie charts that show the proportions of genes in each GO category and sub-category for molecular function, biological process, and cellular component, respectively. The overal counts are given in Fig. S3 and the break-up into categories and sub-categories can be seen in Tables S2A–S2C, for the unique GO terms associated with each cluster, as well as the number of unique genes associated with each cluster. Associated genes are given as percentages of the total number of predicted genes as well as percentages of the number of unique genes associated with each specific GO namespace.

Zinc ion binding and protein kinase activity appear to be the most common categories among the annotated genes, representing 21.54% and 17.64% of the genes annotated for molecular function (Fig. 4A and Table S2A). Interestingly, while protein kinase activity represented 31.38% of the terms out of the 1,144 GO annotations, zinc ion binding was seen in only 3.15% of them, indicating the disproportionately large number of genes involved in this function. While there did not appear to be any such glaring mismatches between the number of genes annotated for a given GO category and the number of GO annotations for that category in the case of biological function (Fig. 4B and Table S2B), unsurprisingly, a similar discord was seen for the cell membrane category in the cellular component GO term (Fig. 4C and Table S2C), attesting to the large number of genes

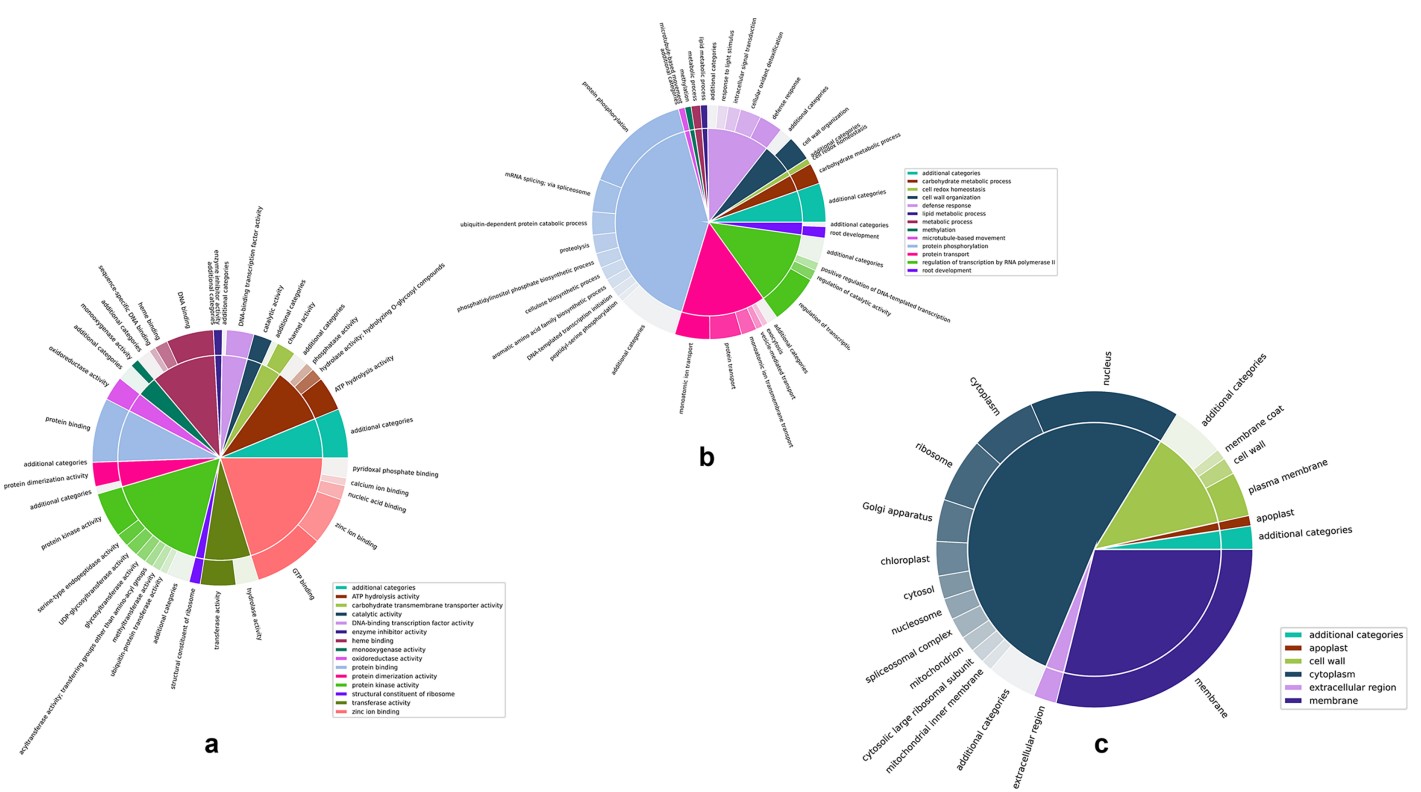

**Figure 4 Two-level pie charts showing the proportion of genes under each category and sub-category for the GO terms.** (A) Molecular function. (B) Biological process, and (C) cellular component.

involved in the membrane to regulate activities such as membrane trafficking (*MartiniEre & Moreau, 2020*).

## CONCLUSIONS

In this study, we used third generation nanopore long-read sequencing technology to assemble a high-quality genome of the orange jewelweed, *Impatiens capensis*. This assembly is expected to contribute toward building a database for comparative genomic analysis aimed at understanding the differences in species-richness and diversification between the genera *Impatiens* and *Hydrocera*. In addition, this new genomic resource should aid the development of cultivars resistant to pathogens in ornamental *Impatiens*. Such a resource will be valuable in continuing to expand our understanding of the biology and evolution of this iconic plant and other species of the genus *Impatiens*, especially in a comparative study with *Hydrocera triflora*.

## ACKNOWLEDGEMENTS

JAB thanks Dr. Vincent P. Richards for bioinformatic support (local server use). SRG thanks Dr. Scott Soby for discussions. We thank the Missouri Botanical Garden, Missouri, USA for generously donating the tissue sample.

### Funding

Sudhindra R. Gadagkar received intramural funding from Midwestern University for this work. The funders had no role in study design, data collection and analysis, decision to publish, or preparation of the manuscript.

### Grant Disclosures

The following grant information was disclosed by the authors:
Midwestern University.

### Competing Interests

The authors declare that they have no competing interests.

### Author Contributions

- Sudhindra R. Gadagkar conceived and designed the experiments, performed the experiments, analyzed the data, prepared figures and/or tables, authored or reviewed drafts of the article, and approved the final draft.
- J. Antonio Baeza analyzed the data, prepared figures and/or tables, authored or reviewed drafts of the article, and approved the final draft.
- Kristina Buss analyzed the data, prepared figures and/or tables, authored or reviewed drafts of the article, and approved the final draft.
- Nate Johnson conceived and designed the experiments, authored or reviewed drafts of the article, and approved the final draft.

### Field Study Permissions

The following information was supplied relating to field study approvals (*i.e.*, approving body and any reference numbers):

Tissue (leaf) samples from a specimen maintained at the Kemper Missouri Native Shade Garden, (Missouri Botanical Garden, Missouri, USA; Accession number 2001-1051) was obtained by one of the authors (SRG) and shipped to Dovetail Genomics (Scotts Valley, CA, USA) for sequencing.

### Data Availability

The genome sequences, assembly, and annotations are available at GenBank:
BioProject: PRJNA875501, BioSample: SAMN30713540.

Kemper Missouri Native Shade Garden:: JAOBSV000000000, Organism: *Impatiens capensis* 2001-1051.

### Supplemental Information

Supplemental information for this article can be found online at http://dx.doi.org/10.7717/peerj.16328#supplemental-information.

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
