# Peer review of "De-novo whole genome assembly of the orange jewelweed, Impatiens capensis Meerb. (Balsaminaceae) using nanopore long-read sequencing"

_PeerJ, doi:10.7717/peerj.16328_

## Round 0.1 · original submission · Major Revisions

Please revise your manuscript according to the comments from the reviewers.

Reviewer 1 ·

Basic reporting

The authors do a fine job describing justifications for sequencing this genome and do a good job describing the methods used to analyze the sequence data produced for the effort; however, I do not see any evidence of the results. From where is the sequenced data available? is it in a database? Is it deposited at NCBI or other third-party resource? I do not see any mention of it within the manuscript.

Experimental design

There are many molecular tools used to analyze the sequence, yet I was not able to obtain any data related to the detailed analysis. There is description of using the dataset to match against other databases; why is the reader not allowed to do the same (there’s nothing to grab onto). For instance a reader might be interested is looking for a particular gene family; how would one start and where would they go?

Validity of the findings

I see that a pseudo-molecule was not able to be constructed, but extensive annotations were done; apparently there are 26921 available; is there even an annotation list. Where would I find out more? The supplementary tables do not suffice in providing details which one would desire to see in a sequencing effort. Since the data is in a partial complete state (in scaffold form); perhaps some comparisons are warranted to describe the completeness of the alignments to other related species available. Sequencing a genome should provide some sort of platform to display your work; this did not. It is more of an announcement, but with no directions provided for the reader to find out more. What may improve the presentation is to present at least some detailed study to accompany the results of this study.

Additional comments

This basically an outline of work that has been done, and once it is ready it should be made available. As an announcement it is just that, but leaves the reader no where to go. I'm interested, but I have nothing to see. Where would one go to follow-up on these results? Will they be placed somewhere that the reader would be familiar with? There may be interesting information here, but it does not fulfill the basic reporting of useful results.

Reviewer 2 ·

Basic reporting

The manuscript describes the whole genome sequencing of Impatiens capensis using nanopore technique. Given that this is a genome report, my comments are as follows:
L26 -aceae itself is addressing the family. Remove redundant words like "family" and "genus" in the text.
L47 the citations are not arranged in an ascending order. Please check throughout the manuscript.
L49 why is Astragalus here?
L50 eventually, this sentence is very long, and it is just to describe Hydrocera as a monotypic genus.
L84-89 I believe few citations would be appropriate to highlight the research gap here.
L91 it was not mentioned that I. capensis is an iconic American species in the Introduction. How iconic is iconic?
L93 the use of congeneric here is inappropriate. sole species is just fine. and genus name should be abbreviated after it was first-mentioned at the start of the write-up.
L105 grammar
L108 if Pac-Bio method is unsuccessful, it would be best to discuss the reason why in the Discussion section. This kind of information is useful for future work too.
L112 What was the insert size based on the library prep protocol?
L113 I do not think that there is a need to report on how long the library was sequenced. Instead, you should inform the readers how much of raw data (Gb) was generated.
L123-131 I supposed these belong to the M&M section
L132 what is ONT?
L141 past tense for your results
L143 parenthesis does not work well here. please check.
L150 I suppose the annotation protocol should be reported under the M&M section, as this is the Result and Discussion section.
L151 (GenSAS; Humann et al. 2019)
L154 parenthesis went wrong here
L156 repeatmasker comes with a reference "Nishimura, D. (2000). RepeatMasker. Biotech Software & Internet Report, 1(1-2), 36-39.". Perhaps there are newer references. Please check.
L167-179 I suppose these protocols should be reported in the M&M section, not in the Results and Discussion section.
L187 This sentence is not helpful to describe what is available in Figure S1 and Table S1.

Some other comments:
1. As a genome report, I would say that this is too little information for a Q2 journal. Generally, I would be looking for at least (1) a list of genes annotated or graphs showing the numbers of genes annotation based on their types, & (2) a diagram of the circular genome map. since it is claimed as a high-quality assembly, a genome map would be possible.
2. Since it is a high-quality assembly, reconstructing a genome-scale phylogenetic tree should not be a problem. Basic phylogenetic information using genome data, although with limited sampling size, will still contribute as a reference to other phylogenetic-related studies. At least, the molecular placement of Impatiens (or Balsaminaceae) based on the full genome sequence can be determined.
3. I am not in a position to perform English check; however, I do detect some long, winding sentence, as well as grammar mistake in the manuscript. Please check on that too.
4. Lastly, as mentioned in the rebuttal letter to the previous editor, the authors are reporting a genome data. I would say that all genomic data are important for a reason and should not be rejected due to lack of a series of comparative analyses. However, I would highly suggest the authors to include basic analyses in this report (i.e. types of genes annotated, phylogenetic analysis, etc.). Eventually, I would let the editor to decide whether this manuscript should be accepted for publication in PeerJ based on the weight of the content reported.

Experimental design

please refer abovementioned comments

Validity of the findings

please refer abovementioned comments

Additional comments

please refer abovementioned comments

---

## Round 0.2 · Minor Revisions

Please take the comments from reviewers into account when revising your manuscript.

Reviewer 1 ·

Basic reporting

The authors do a good job of reporting the basic problem and justifications for the sequencing project; however, within the text of the manuscript there is no mention of how the summaries of the data can be connected to sequence data. The only pointer is to the SRA data which is basically the raw data. It should not be put on the reader to jump through the protocols to replicate the work. The was a mention of data located at Figshare (mentioned only in Rebuttal), but there is no connection to it from within the resource of text in the manuscript.

Experimental design

The experimental design is fine as it was farmed out to a sequencing resource. The methodology was well described but it might be a bit clearer if one could distinguish how much was corporately generated as that from the research laboratory.

Validity of the findings

The validity of the finding really cannot be determined without downloading the Figshare data resource which is not mentioned within the text of the manuscript; yes it’s there, but not discussed of referenced! The Venn diagrams for gene annotations cannot be connected to sequence data; however, it might be in one of the GFF files which is not mentioned within the context of the manuscript. Usually an accompanying spreadsheet would accompany the Venn diagram to itemize which sequences were in each categorization group or pie chart.

Additional comments

The data might actually be there, but the authors to be more clear in itemizing how the reader can follow the data on their own rather than taking for granted the summarized data. I suggest a major revision, mainly to add clarity.

Reviewer 2 ·

Basic reporting

I have read through the whole manuscript, as well as the changes made according to the comments I have brought forward in my previous review. Fair enough, the authors have definitely answered to most of my comments, and also gave me good reasons for those that are not relevant for changes. Overall, I am impressed and accept the manuscript at its current form. In my opinion, the finding of this study is valuable and should not be excluded from publication at PeerJ.

Experimental design

no comment

Validity of the findings

no comment

Additional comments

no comment

---

## Round 0.3 · accepted · Accept

The revised version is acceptable for publication.